:Φᐧ PLOS ONE

# ViraMiner: Deep learning on raw DNA sequences for identifying viral genomes in human samples

Ardi Tampuu[1]☉, Zurab Bzhalava[2]☉, Joakim Dillner[2,3], Raul Vicente[1]*

**1** Computational Neuroscience Lab, Institute of Computer Science, University of Tartu, Tartu, Estonia, **2** Department of Laboratory Medicine, Karolinska Institutet, Stockholm, Sweden, **3** Karolinska University Laboratory, Karolinska University Hospital, Stockholm, Sweden

☉ These authors contributed equally to this work.
* raul.vicente.zafra@ut.ee

**Data Availability Statement:** All necessary data files are available from the GitHub repository associated with this work: https://github.com/NeuroCSUT/ViraMiner. Also all code is available in there.

## Abstract

Despite its clinical importance, detection of highly divergent or yet unknown viruses is a major challenge. When human samples are sequenced, conventional alignments classify many assembled contigs as "unknown" since many of the sequences are not similar to known genomes. In this work, we developed ViraMiner, a deep learning-based method to identify viruses in various human biospecimens. ViraMiner contains two branches of Convolutional Neural Networks designed to detect both patterns and pattern-frequencies on raw metagenomics contigs. The training dataset included sequences obtained from 19 metagenomic experiments which were analyzed and labeled by BLAST. The model achieves significantly improved accuracy compared to other machine learning methods for viral genome classification. Using 300 bp contigs ViraMiner achieves 0.923 area under the ROC curve. To our knowledge, this is the first machine learning methodology that can detect the presence of viral sequences among raw metagenomic contigs from diverse human samples. We suggest that the proposed model captures different types of information of genome composition, and can be used as a recommendation system to further investigate sequences labeled as "unknown" by conventional alignment methods. Exploring these highly-divergent viruses, in turn, can enhance our knowledge of infectious causes of diseases.

## Introduction

The human virome is the collection of all viruses that reside in and on the human body. Many different viruses are present in human samples and their composition appears to be different in diseased individuals [1, 2]. Despite its clinical importance, its full impact on human health is not fully understood [3, 4] and the detection and classification of human viruses represents a major challenge.

Current metagenomic studies detect many novel viruses, which indicates that only a small part of human viruses has been discovered and many others are yet to be reported [5–10]. Studies report epidemiological indications that there may exist undiscovered pathogens. For

**Funding:** ZB and JD were supported by three sources: 1) awarded to JD: Swedish Foundation for Strategic Research, Proj. no RB13-0011, https://strategiska.se/en/; 2) awarded to JD: NordForsk, Proj no 62721, https://www.nordforsk.org/en?set_language=en; 3) awarded to JD: Swedish Research Council, Proj no 2017-01841_3, https://www.vr.se/english.html. AT and RV were supported by Estonian Research Council, project number PUT 1476 (https://www.etis.ee/Portal/Projects/Display/52ed4301-f2ef-4364-9770-397e31936f93?lang=ENG) and Estonian Centre of Excellence in IT (EXCITE) project number TK148 (https://www.etis.ee/Portal/Projects/Display/fd0aeffa-a7d3-4191-b468-0f44aa2847af?lang=ENG). The funders had no role in study design, data collection and analysis, decision to publish, or preparation of the manuscript.

**Competing interests:** The authors have declared that no competing interests exist.

example, there is correlative evidence suggesting that viruses may be involved in the development of autoimmune diseases such as diabetes [11] and multiple sclerosis [12].

Next Generation Sequencing (NGS) technologies provide a powerful tool to obtain directly the DNA sequences present in clinical samples without any prior knowledge of the biospecimens [13]. The term metagenomics implies complete sequencing and recovering of all non-human genetic material that might be present in a human sample and accordingly, analytical techniques for virus discovery are commonly applied to metagenomic samples [5, 7, 8, 14–19].

Currently, the detection of potential viral genomes in human biospecimens is usually performed by NCBI BLAST, which implements alignment-based classification where sequences are aligned to known genomes from public databases and then estimates how much similarity they share. However, metagenomic samples might contain a large number of highly divergent viruses that have no homologs at all among known genomes. As a consequence, many sequences generated from NGS technologies are classified as "unknown" by BLAST [5, 18]. Another commonly used algorithm for viral discovery in metagenomic samples is HMMER3 [20], which applies profile Hidden Markov Models (pHMM) in relation to the vFams [21] database of viral family of proteins, that have been built by multiple sequence alignments from all viral proteins in RefSeq. With this method, sequences are compared to entire viral families which enables the algorithm to be more effective in detecting distant homologs [22]. However, this pipeline needs a reference database which is a drawback while conducting the analysis of identification of highly divergent viruses. A different approach consists of using machine learning techniques to learn from examples to classify viral genomes and to generalize to novel samples. In particular, several machine learning models for detecting viruses in metagenomic data have been already published [23–25], but none of them were trained nor tested to identify viruses in different human biospecimens.

In this study, we have developed ViraMiner, a methodology which uses Convolutional Neural Networks (CNN) on raw metagenomic contigs to identify potential viral sequences across different human samples. The architecture of ViraMiner builds on top of the CNN architecture of Ren [25], adding major extensions for it to be more effective on this particular classification problem. For training the model we used 19 metagenomic experiments originating from sample types such as skin, serum, and condylomata. The model achieves a significantly improved accuracy compared to other existing methods for virus identification in metagenomic samples. To our knowledge, the proposed model is the first methodology that can detect the presence of viruses on raw metagenomic contigs from various human biospecimens. A pre-trained model ready to be used is also publicly available on GitHub (https://github.com/NeuroCSUT/ViraMiner).

## Results

First we will describe the overall architecture of the proposed model (ViraMiner) for the detection of viral genomes from assembled metagenomic contigs (see Methods for a detailed description of the architecture, its inputs and outputs and details on training procedure). In the following sections, performance measures of the model and multiple baselines are reported. Finally, an evaluation of the model when applied to novel metagenomic experiments is given.

### *ViraMiner* architecture

In this work we developed ViraMiner, a CNN-based method for detecting viral contigs in human metagenomic datasets. As illustrated in Fig 1, ViraMiner receives a raw DNA sequence as input and outputs a single value that indicates the likelihood of this sequence being of viral

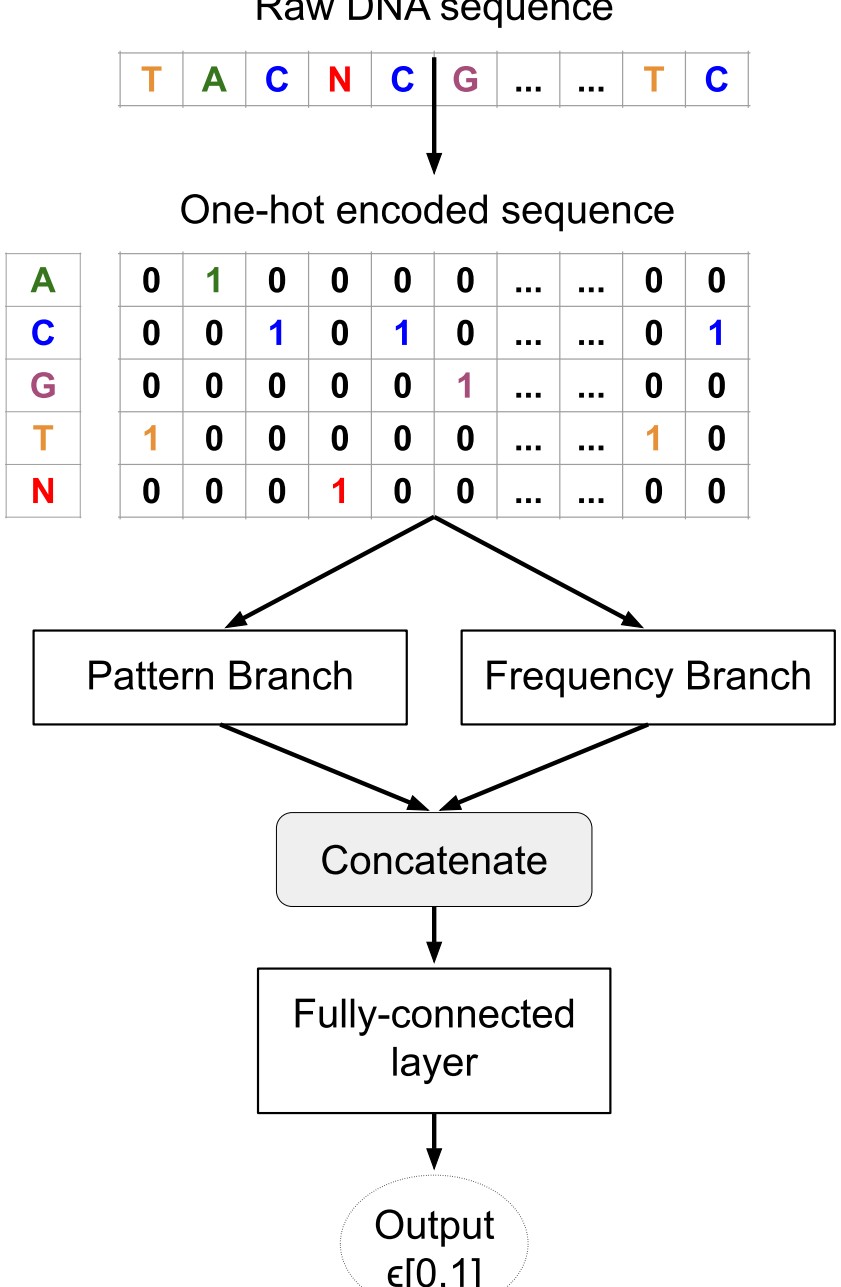

**Fig 1. ViraMiner architecture.** ViraMiner architecture takes as input the raw sequences in one-hot encoded form. The raw sequences are then processed by two different convolutional branches (pattern and frequency branch). The architecture of the branches is shown in detail in Methods and Material. The outputs of the branches (two 1D vectors) are concatenated. This concatenated vector is all-to-one connected (fully-connected) with the output node. The output node passes the weighted sum of its inputs through sigmoid activation function resulting in a value restricted to range [0, 1]. This value reflects the likelihood of the sequence belonging to virus class.

origin. In ViraMiner we apply convolutional layers with learnable filters straight on the raw sequences. This way the model has access to all information and can learn by itself from labeled examples which features are important to extract (which information to keep). To maximize the model's ability to extract relevant features, the ViraMiner model has two different convolutional branches (see Fig 1). Intuitively, one of the branches (pattern branch) returns how well certain patterns were maximally matched along the DNA sequence. In this branch the convolutional layer is followed by *global max pooling* that outputs only the maximal value for each convolutional filter. The other (frequency branch) returns information about pattern frequencies. In this branch the convolutional layer is followed by *global average pooling* that yields the average value for each convolutional filter. The outputs of these branches are merged and the output node is all-to-one connected (fully-connected) with them. Sigmoid activation function is applied in this node, transforming the weighted sum of inputs to a probability (into range [0, 1]).

## Identifying viral genomes in human samples

The proposed models were trained on metagenomic assembled contigs from 19 different experiments, which in turn were sequenced from different human sample types (serum, skin, condylomata, . . .). These contigs were combined, shuffled, and partitioned into training (80%), validation (10%), and testing (10%) sets. Models were trained using the training set and the best model was selected according to validation set performance (see "Hyperparameter search and testing" subsection in Methods). The test set was used only for the final evaluation of the best models.

The proposed architecture achieved outstanding test performance on this human metagenomic data. In Fig 2 we show the Receiver Operating Characteristic (ROC) curve of the best ViraMiner model with an area under the curve (AUROC) of 0.923.

Given the huge class imbalance of this type of datasets (e.g. in the studied human metagenomics dataset there were only 2% of viral contigs) it is important to evaluate precision and recall measures for both classes separately. Overall measures are hardly informative—a trivial model classifying everything as non-virus would achieve 98% overall precision and 98% overall recall, but it would nevertheless be useless for detecting viral genomes. Thus, next we report the precision and recall of the virus class instead of global performance. Note also that to measure particular recall and precision values, one would need to establish a classification threshold. The higher the threshold is set, the higher the precision gets but at the expense of recall. On the other hand, setting a lower threshold would increase recall but decrease precision. Fig 3 represents the tradeoff between precision and recall for the virus class. The dotted lines indicate that ViraMiner could achieve 90% precision with 32% recall. With a stricter threshold, one can obtain 95% precision and 24% recall. Inversely, if one wishes to increase recall and accept relatively more false positives, the model can reach 41% recall with 70% precision.

## Baselines comparison

**Contributions of each CNN branch separately.** The model training proceeds by training the Pattern and Frequency branches separately. Consequently, we can also report the performance of each branch alone. With hyperparameters selected to maximize area under the ROC curve (AUROC) on validation set, on test set the Pattern branch achieves AUROC 0.905. In the same conditions, the Frequency branch achieves a test AUROC of 0.917. This shows that the filters of each branch result in a relatively high accuracy despite their different pooling operations, and their combination results in an even higher AUROC score (0.923). The stepwise training of ViraMiner architecture is important to avoid overfitting. When optimizing

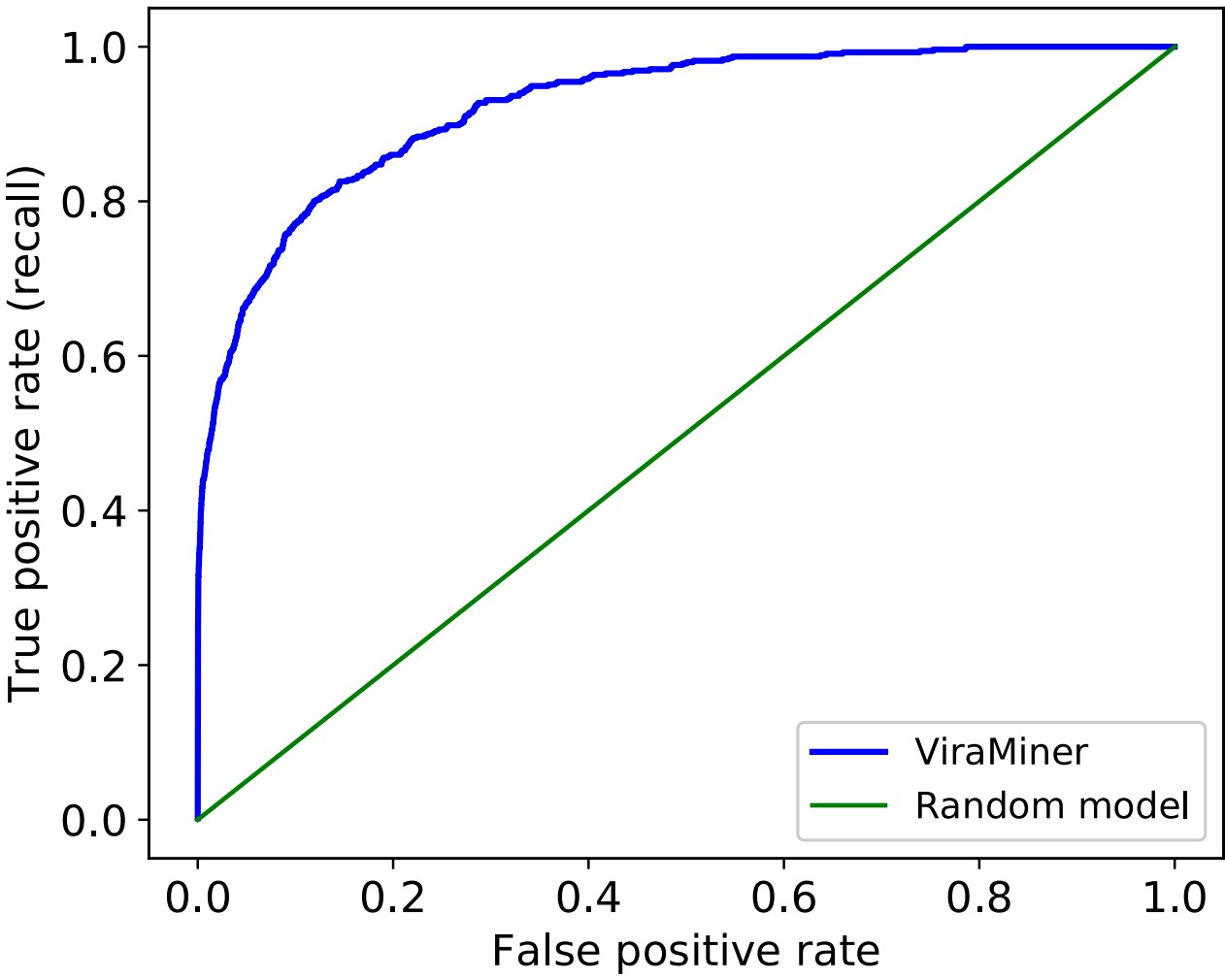

**Fig 2. ROC curve of ViraMiner model on the testing dataset.** The blue line on the figure represents the performance of ViraMiner (AUROC: 0.923). For comparison, the green line depicts the performance of a random model.

altogether the 2 million parameters in a ViraMiner model at the same time (both branches trained simultaneously), the performance on test set decreases. The test AUROC of such end-to-end trained model is only 0.896.

**K-mer models.** Most of the previous machine learning models for classifying metagenomic datasets are based on k-mer counting [24, 26, 27]. To compare against the performance of such methods, we also extracted k-mers from the investigated dataset and trained Random Forest (RF) classifiers on the extracted values, while keeping the same data partitioning as above. RF is a competitive machine learning algorithm for non-linearly separable classes and it has already been used on this type of datasets [23, 28]. The best test performance with RF models was achieved with 6-mers and it produced test AUROC 0.875 (Fig 4). RF performances on 3- 4- 5- and 7-mers were 0.867, 0.872, 0.873, and 0.869 respectively (Fig 5).

For completeness, we also trained a RF model on raw metagenomic contigs (one-hot encoded and flattened), that is the same input used to ViraMiner, and hence without the

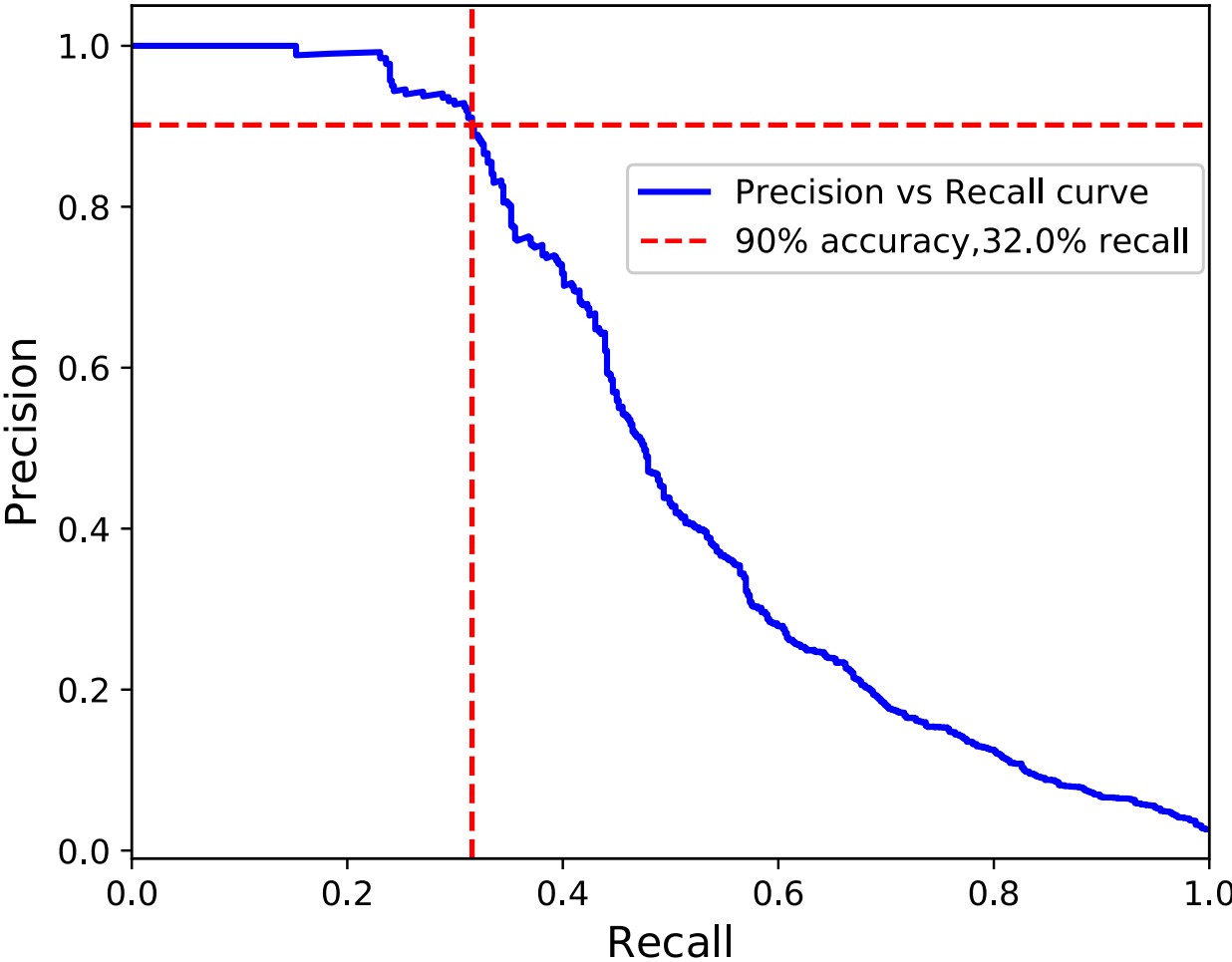

**Fig 3. The trade-off between precision and recall for classifying viral genomes.** A genome is labeled as virus if the output value is higher than a certain threshold. The blue line represents the trade-off. The crossing point of the vertical and horizontal red dotted lines indicates that the model can achieve 90% precision with 32% recall.

manual extraction of different k-mers counts. The test performance of such RF model was very close to a random model (AUROC: 0.573).

The overall comparison of test performances of models designed in this study is summarized in Fig 5.

**ViraMiner as a recommendation system.** In here, we exemplify the use of ViraMiner a recommendation system. We apply the model to the entire test set and order the test sequences according to the likelihood of being viral returned by the model. We then see how many of the top N most likely sequences are indeed viral.

In particular, we compared Positive Predictive Value (PPV) of ViraMiner with PPV of the best RF (6-mer) model. Top 20 test set sequences most likely to be viruses according to both models were indeed viruses. However, there was a clear difference between confidence rates: confidence of ViraMiner for its top 20th virus was 99% while RF yielded only 67% for the corresponding 20th poisiton. For the top 200 sequences, ViraMiner achieved 88% PPV whereas

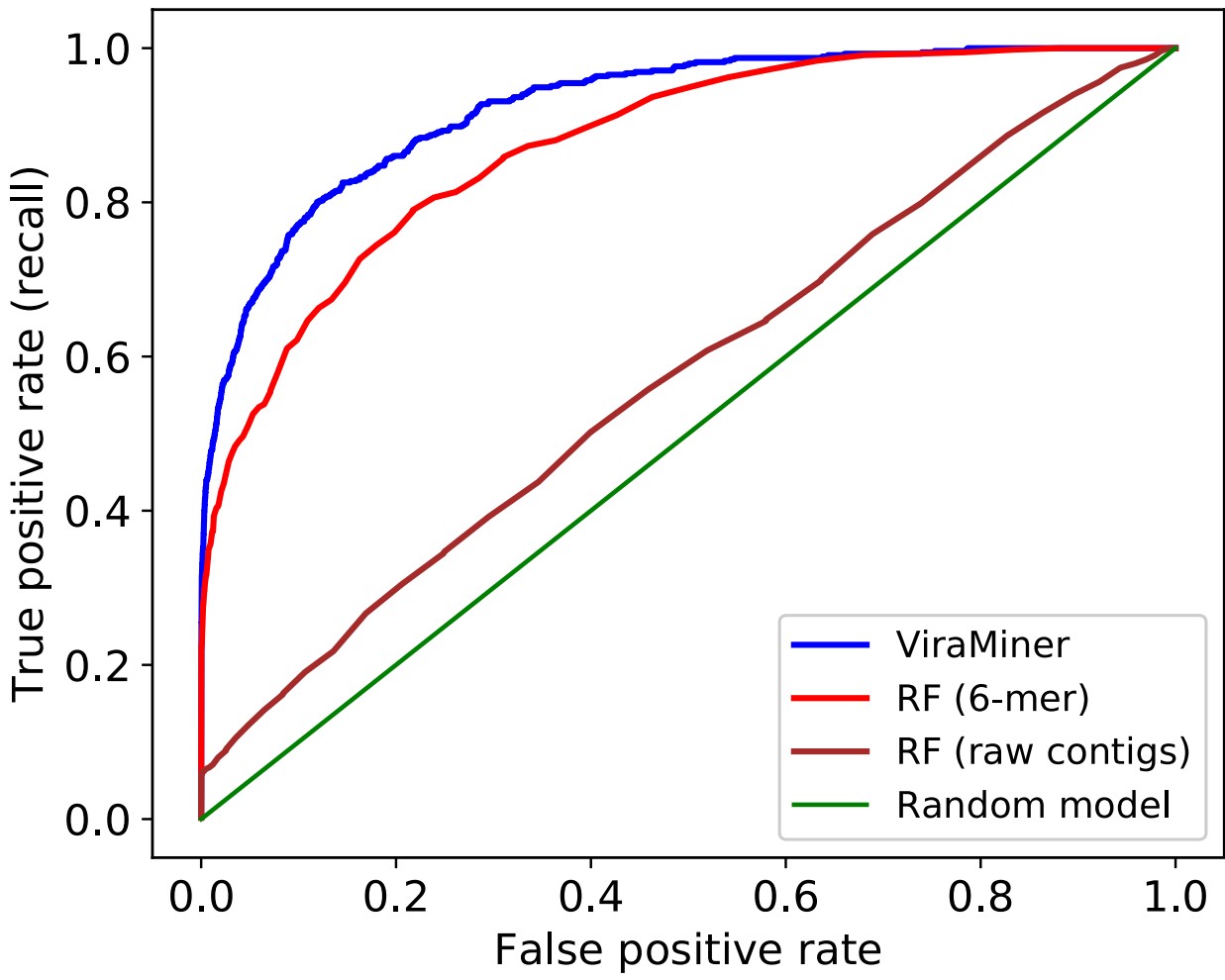

**Fig 4. Comparison with baseline models.** The blue line depicts the performance of ViraMiner (AUROC: 0.923); the red line corresponds to the best of Random Forest on k-mers models (using 6-mers); the brown line, shows RF performance on raw metagenomic contigs.

RF reached 75%. Pattern and Frequency branches for the same number of top sequences output 79% and 83% PPVs respectively.

### Generalization to unseen datasets

**Performance on previously unseen sequencing experiments.** In this section we describe the performance of the proposed model on novel metagenomic experiments. We retrain the model with the same hyperparameters as above (no new search is performed), but we remove one sequencing experiment completely from the train and validation sets and use it as the test set. Notice that the training procedure remains absolutely the same—we train the models (branches first, then the full model) until validation AUROC stops increasing. Afterwards, the model is applied to the test set—the left-out data originating from a novel sequencing experiment.

We repeat this leave-one-experiment-out procedure for all the 5 metagenomic datasets originating from serum sample type. In here, we selected to use serum type because among

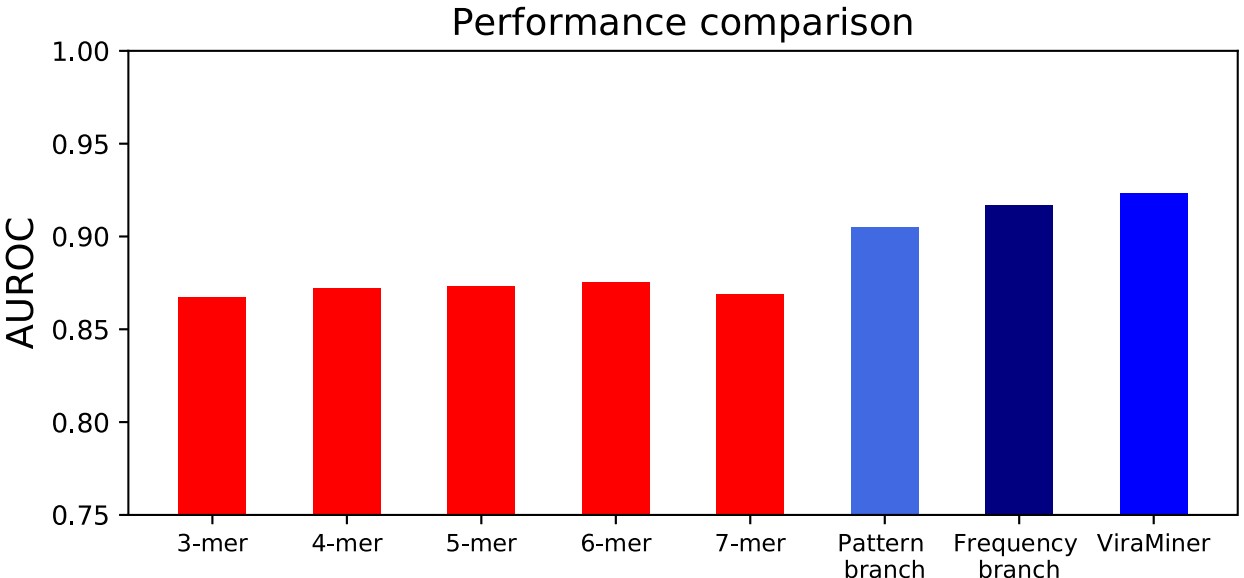

**Fig 5. Performance comparison of models designed in this study.** Models based on k-mers (red bars) were trained using Random Forest and the best performance was achieved with 6-mers with test AUROC value 0.875. Convolutional Neural Networks (blue bars) use the raw sequence as input and outperform RF models. Pattern and Frequency branches yield performances above 0.9 AUROC. ViraMiner uses a combination of both branches and achieves the highest test AUROC value (0.923) out of all models.

our datasets largest number of experiments and largest number of samples originate from this sample type. Thus, in total we train 5 ViraMiner models, each time leaving a different serum dataset out for testing. We do this to show how much the model's generalization ability varies dataset to dataset. The test performances of these models are given in Table 1. These results show that ViraMiner models generalize well to all 5 serum datasets.

The ViraMiner model achieved good test performance on all 5 serum datasets, with AUROCs ranging from 0.86 to 0.96. To provide a unique measure, we merge the predictions for all five sets and redraw the ROC curve. The area under this micro-averaged ROC curve across the five test sets is 0.94.

We also investigated whether training a ViraMiner model on data originating from only serum metagenomic datasets would improve the performance compared to a model trained using a variety of metagenomic datasets (as above). For this purpose, we trained ViraMiner model on assembled contigs originating only from four serum experiments and tested on the 5th serum dataset. Again, we repeated the leave-out training procedure five times, so each dataset was left aside as test set once. With this reduced but more specific training set, the test area under micro-averaged ROC curve was 0.91. This is a noticeable decrease compared to 0.94 AUROC achieved when using all available training data. In our case using all available data points was useful even though many of these data points came from completely different sample types such as skin, cervix tissue, prostate secretion, etc.

**Table 1. ViraMiner performance on unseen serum metagenomic experiments.** The number of viral samples in these experiments varies from 22 to 732.

| Left-out experiment | 2011_G5 | 2014_G1 | 2014_G5 | 2014_G6 | 2014_G7 | Micro-average |
|---|---|---|---|---|---|---|
| test AUROC | 0.95 | 0.89 | 0.96 | 0.92 | 0.86 | 0.94 |

**Performance on unseen viral class.**   In addition, we investigated whether ViraMiner is able to identify viruses from a viral class that it had not encountered during training. For this purpose, we trained and validated the ViraMiner model on a dataset where all anelloviruses were removed. We used the same hyperparameters (layer sizes) with the same step-wise training strategy—i.e. no new hyperparameter search was performed. The test set contained non-viral samples (26296 contigs) and the anelloviruses (1348 contigs). In this setting, from the model's point of view anelloviruses are a completely unknown and unseen type of viruses.

On this test set, the frequency branch alone achieves the best result, with 0.755 AUROC. This is clearly above random performance and translates into getting 11 of the top 20 predictions correct on a test set with 5% prevalence. This means that even when dealing with viral sequences that are very distant from our training (and validation) samples, using ViraMiner as recommendation system strongly increases the chances of identifying viruses.

**Performance on simulated data.**   To confirm our model's high efficiency is not due to unknown biases in our datasets, we also used data simulated with ART sequencing read simulator [29] (see Methods for simulation procedure and parameters). This resulted in slightly fewer sequences (210 000 vs 260 000) but a higher prevalence of viral sequences (roughly 10%).

First we tested the ViraMiner model trained on data from our 19 metagenomic experiments (trained on real data) on this simulated dataset. The resulting AUROC was 0.751. This performance is similar to what was reported in the previous subsection for unseen viral class. Indeed, simulated data contained samples from randomly selected viral species, including those not present in the real data (the training set).

Secondly, we also retrained the model on simulated data. As the amount of data was similar we used the exact same hyperparameters and training procedure. ViraMiner model was trained on 80% of the simulated data, validated on 10% and tested on 10%. The resulting test AUROC was 0.921, whereas the branches separately achieved 0.914 and 0.928 for pattern and frequency branch respectively. These results indicate that the methods (architecture, training procedure) perform equally well on simulated and real data.

## Discussion

In this work, we proposed ViraMiner, a CNN-based method to detect viruses in diverse human biospecimens. The tool includes two different branches of CNN: a Pattern branch with convolution+max operator and a Frequency branch with convolution+average operator.

Firstly, we trained the model on assembled metagenomic contigs originating from 19 metagenomic experiments which were merged and partitioned into training, validation and testing sets. The model achieves 0.923 test area under the ROC curve. This is a significant improvement compared to a previous tool ([28], reaching 0.79 AUROC) that was designed using relative synonymous codon usage (RSCU) and trained on data originating from the same metagenomic experiments. We are confident to conclude that using raw metagenomic contigs with CNN can extract much more information for the problem of viral classification. Furthermore, another advantage of ViraMiner is that a sequence is not required to have an Open Reading Frame (ORF) which is a central requirement for the models based on codon usage.

VirFinder and DeepVirFinder (DVF) are other similar tools available for detecting viruses in metagenomic sequencing datasets [24, 25]. The former is based on k-mers while the latter applies CNN on raw DNA sequences. These tools, however, were specifically trained to identify viruses infecting prokaryotic organisms whereas we trained the model to detect any virus that might appear in multiple human samples. The architecture of ViraMiner builds on top of the CNN architecture of DeepVirFinder, adding major extensions for it to be more effective on this particular classification problem. Indeed, the newly introduced Frequency model

separately performed better than the DVF architecture (referred to as Pattern model in Results). Considering these two studies, DVF and the present article, it is clear that such CNN architectures work very well on viral genome classification.

We also explored whether the proposed CNN architecture was able to extract more informative features from raw metagenomic contigs than to just count k-mers on them. We counted 3-,4-,5-,6- and 7-mers on the contigs, trained Random Forest and compared the results with our model. Since ViraMiner produced a higher accuracy (Fig 4) we can deduce that the architecture extracts more complex features of genome compositions that can detect patterns that cannot be identified with just counting k-mers.

The most important criteria that ViraMiner had to satisfy, however, was to generalize its classification abilities on completely new and unseen metagenomic experiments from which the model had not seen any data point. To this end, not only did we test ViraMiner on one specific experiment but also one specific sample type. In our dataset, the largest number (five) of experiments came from serum and therefore, we tested if the model could generalize on all these five experiments. ViraMiner was retrained five times with the same hyperparameters but each time data from one experiment was left out and used as the test set. The model produced slightly different test AUROC values for each of these datasets which gave an average 0.94 area under ROC curve.

The method's predictive ability was also confirmed on simulated data and on a test dataset containing only viruses from an "unknown" viral family that was left out during training and validation. In these additional verification tasks, no hyper-parameter search was performed and the numeric results can probably be improved upon when optimizing for these tasks in particular.

Certainly, at this stage ViraMiner or other deep learning based methods cannot replace BLAST or HMMER3 partly because of the limited number of data points in the training dataset. However, it is important to note that ViraMiner does not have a database to rely on and must thus capture a different kind of information of genome composition than alignment based methods. Considering the results described above, the proposed method can be used as a recommendation system to further research "unknown" sequences labeled by the conventional methods among which highly divergent viruses may be hidden. Pre-trained models of ViraMiner are publicly available at https://github.com/NeuroCSUT/ViraMiner. We hope that this and future methods can discover biological attributes of viral genome to predict and reveal viruses, and in turn, help focus our exploration of infectious causes in diseases.

## Methods and materials

### Samples and sequencing types

The metagenomic sequences in this work were generated using Next Generation Sequencing platforms such as the NextSeq, Miseq, and HiSeq (Illumina) as described by the manufacturer. The dataset was derived from human samples belonging to different patient groups that have been described in detail in [6–8, 30–32]. The goal of those analyses was to detect viral genomes or other microorganisms in diseased individuals or in their matched control subjects.

### Bioinformatics

All the sequencing experiments were processed and analyzed using a benchmarked bioinformatics workflow [33]. To summarize, we start analysis with quality checking where reads are filtered based on their Phred quality score. Afterwards, quality checked reads that align to human, bacterial, phage and vector DNA with >95% identity over 75% of their length are subtracted from further analysis using BWA-MEM [34]. The reads that are left are normalized

and then assembled using the IDBA-UD [35], Trinity [36], SOAPdenovo, and SOAPdenovo-Trans [37] assemblers. The assembled contigs are then subjected to taxonomic classification using BLAST. The code of the pipeline is available on GitHub (https://github.com/NGSeq/ViraPipe and https://github.com/NIASC/VirusMeta).

## Data processing and labeling

The training dataset included 19 different NGS experiments that were analyzed and labeled by PCJ-BLAST [38] after applying the de novo genome assembly algorithms. Parameters for PCJ-BLAST were as follows: type of algorithm = Blastn; nucleotide match reward = 1; nucleotide mismatch penalty = 1; cost to open a gap = 0; cost to extend a gap = 2; e-value $\leq$ e−4. All assembled contigs that were labeled by this bioinformatics workflow were merged to train several machine learning algorithms.

To extract input dataset for Convolutional Neural Networks, labeled sequences were divided into equal pieces (300bp and 500bp), each one of them was labeled as the original sequence and remaining nucleotides at the end of the contigs were removed from the further analysis. For example, according to this processing 650 bp viral contig would produce two equal 300bp viral sequences and the last 50 nucleotides would be removed from the input dataset. Initially, we generated two training datasets based on sequence lengths: first with 300bp and the second with 500bp. After noticing that 300bp contigs produced significantly better results we continued working only on this dataset and dropped the other. Furthermore, all contigs that contained even one "N" letter (reference to any nucleotide) were removed from the further analysis.

The most common genes found in the viral sequences in our metagenomic datasets are listed in S1 Table. The most common viral classes are listed in S2 Table.

For computing baselines, we also counted k-mers in the processed dataset. Given that extracting k-mers becomes more and more computationally expensive as k increases, we conducted distributed and parallel computing of k-mers by using Hadoop (https://hadoop.apache.org) and Spark [39]. The Spark code (available under https://github.com/NIASC/ViraPipeV2) outputs $4^k \times l$ matrix where $l$ is number of rows from an input dataset.

## Machine learning methods

**Convolutional neural networks (CNNs).**   Convolutional neural networks (CNNs) are a type of feed-forward neural networks [40, 41]. In addition to fully-connected layers that have all-to-all connections, CNNs by definition also contain convolutional layers. Convolutional layers have partial connectivity and use the same parameters repeatedly [40]. In supervised learning settings, CNNs learn to perform a task by observing input-output pairs. The learning is achieved by minimizing the error (between model's output and the known true output) via gradient descent [40, 42].

CNNs are most widely used in image processing [40, 43], but have been successfully applied in various fields [44, 45], including analysis of biological sequences such as DNA and amino acid sequences [46–48]. The convolutional layers treat their inputs as multidimensional arrays—an image is treated as a 2D array with 3 color channels, while a DNA sequence is seen as a 1D array with one channel per possible nucleotide value. In the present work we use sequences of length 300 and consider 5 possible values (ATGCN) at each position, corresponding to a 1D sequence of length 300 with 5 channels (as depicted in Fig 1). A convolutional layer is characterized by a set of learnable filters that are convolved along the input DNA sequence (or along the width and height of an image). Filters are smaller than the input and can be applied at multiple locations along the input. At each position where the filter is applied,

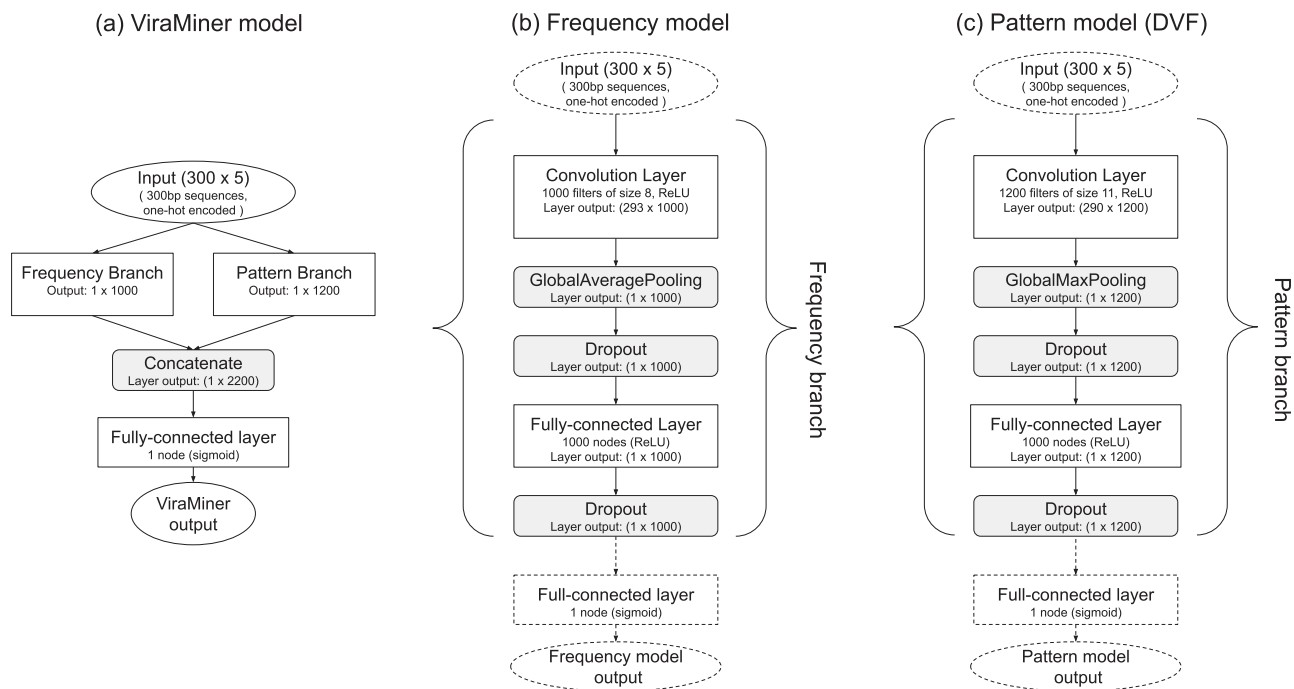

**Fig 6. ViraMiner architecture.** (a) ViraMiner full model, with the branches not expanded. (b) Architecture of the Frequency model with best performing layer sizes shown. The dotted parts will be discarded when using the pre-trained parameter values of this model as the Frequency Branch in the full model. (c) Architecture of the Pattern model (similar to DeepVirFinder) with best performing layer sizes shown. The dotted parts will be discarded when using the pre-trained parameter values of this model as the Pattern Branch in the full model.

a dot product between the weights of the filter and the local input values is computed. The dot products resulting from applying the same filter to different positions along the sequence form a feature map. A convolutional layer usually applies many filters and hence its output is a set of feature maps, one per filter. On this output, further computations can be performed such as applying an activation function, pooling or adding further layers. The parameter values in the filters, like all other learnable parameters in the model, are optimized by the gradient descent algorithm.

**CNN architectures for predicting the viral nature of sequences.** In this work we use a convolutional neural network called ViraMiner to predict whether a given DNA sequence is of a viral origin or not. This model contains two convolutional branches that contribute different types of information to the final fully-connected decision-making layer (Fig 6a).

This model is partly based on the DeepVirFinder (DVF) architecture by Ren et al [24]. The DVF model is a Convolutional Neural Network (CNN) that applies the *max operator* on the outputs of each convolutional filter (on each feature map). Only the one maximal activation value per filter is passed on to the next layers. All other information, such as how often a good match was found, is lost. This architecture is illustrated in Fig 6c.

In this work we suggest that a convolutional layer followed by *average operator* instead of *max operator* provides important information about the frequency of patterns, information that is otherwise lost. In such a case, while losing information about the maximal activation (best match), we gain information about frequency—the average cannot be high if only a few good matches were found. In previous work the authors of DVF and the authors of the current article have shown that methods based on pattern frequency (k-mer counts, relative synonymous codon usage) are effective in separating viral samples from non-viral ones [24, 28].

Using *convolution + average* is a natural extension to these pattern counting-based models. This results in the architecture in Fig 6b.

To clarify, we do not claim that using *convolution+max* is without merit, it simply captures a different type of information. In fact, in our ViraMiner model we combine feature maps processed by averaging and by max operators. This allows the model to base its decisions on both pattern-matching and pattern-frequency information. The ViraMiner detailed architecture is shown in Fig 6a where Frequency Branch and Pattern Branch refer to the architectures in Fig 6b and 6c, without the input and output layers.

**Training.**   It is important to note that Pattern and Frequency models can be used as separate models and trained independently. In the first step of our training scheme we train these two models, then remove the output layers and use the middle layers as Frequency and Pattern branches in the full ViraMiner model. In the second step we optimize only the parameters in the final layer of the ViraMiner architecture, leaving the weight and bias values in the branches unchanged, exactly as they were in the independent models. Notice that in this last step of learning, values of only roughly two thousand parameters are changed.

This *step-wise* training procedure helps reduce overfitting compared to simpler approaches. The simplest alternative training strategy consists in randomly initializing and optimizing all the parameters (both branches and the final layer) at the same time. This standard *end-to-end* approach allows the model to freely use all of its capacity to minimize the error in training data points. However, too much capacity might lead to lower generalization ability. Indeed, with this training procedure ViraMiner model strongly overfits and performs significantly worse on unseen data. *Fine-tuning* is another alternative training procedure where the branches are initialized with pre-trained values, but the values are not frozen. This means that when training the full ViraMiner model both the final layer and the branches are optimized. The re-optimization of pre-trained weights gives the name *fine-tuning*. Similarly to *end-to-end* training we see overfitting in this intermediate training procedure. Therefore, the models in the Results section are obtained with *step-wise* training procedure, rather than *end-to-end* or *fine-tuning*.

**Hyperparameter search and testing.**   We merged and shuffled the data originating from 19 metagenomics sequencing experiments and divided it into training (80%), validation (10%) and test sets (10%). We then performed an extensive hyperparameter search for the Pattern and Frequency models separately. For both architectures, hundreds of variants were trained using the 80% of data in training set. The performance of these models was measured by AUROC on validation set. The models were trained until validation AUROC had not increased for 6 consecutive epochs, but not longer than 30 epochs. After each epoch the model was saved only if validation AUROC had increased. Adam optimizer [49] and batch size 128 were used in all cases.

The initial hyperparameter search scanned the following parameters:

- Filter size: in range from 4 to 40 with step of 4.

- Learning rate: 0.01, 0.001 and 0.0001. Each value was tried with and without learning rate decay, which multiplies learning rate by 0.5 after each 5 epochs.

- Layer sizes (applies both to number of filters in convolutional layers and to number of nodes in fully connected layer): 1000, 1200 or 1500

- Dropout probability: 0.1 or 0.5

In a second stage of hyperparameter search, each possible filter size value (with step of 1) was scanned in the region of interest determined by initial scan (8 to 16 for Frequency model;

6 to 14 for Pattern model). For other hyperparameters, some values were left out from the second stage as they clearly worsened the performance.

When the extensive hyperparameter scan was completed, the best Pattern model and best Frequency model (as per validation AUROC) were selected and used to initialize the branches in the ViraMiner architecture. We then trained the ViraMiner model with different learning rates, learning rate decays and with/without fine-tuning the branches and again selected the best model according to validation AUROC.

Having selected best models according to validation performance, we finally evaluated the best Pattern, best Frequency and best ViraMiner model on the test set which had been left unused up until this point.

**Baselines.**   Using 80/10/10 split, we also trained two types of baseline models. First of all, using k-mer counts extracted from the contigs as input, we trained Random Forest models with 1000 trees (all other parameters as default in scikit-learn Python package [50]) for k in range from 3 to 7. In the Results section, we report each model's test AUROC.

Secondly, we also trained a Random Forest model directly on the 300bp sequences, without any feature extraction. For that we one-hot encoded the sequences and flattened them, resulting in 1200 inputs, 4 per each position in the sequence. Notice that such model is not position invariant—a shift of the sequence by just one base pair would completely change the output.

**Simulated data.**   Simulated pair-ended reads for this study were generated with ART simulator [29]. The reference file for the simulation included 30 thousand genomes from Gen-Bank consisting 2870 viral and 27 130 cellular organisms (mammals, plants, vertebrates and bacterias). Parameters for the simulator were as follows: -ss MSv1 -p -l 150 -f 20 -m 200 -s 10. The simulation produced appproximately 13 million paired reads which in turn were assembled into 46 498 contigs by using IDBA-UD assembly [35]. Afterwards, we applied the same data processing and labeling pipeline as described for the 19 metagenomic datasets.

## Code availability

All code, for ViraMiner model and for baselines, was written using Python2.7. For neural networks, Keras library was used [51]. All code is publicly available on GitHub https://github.com/NeuroCSUT/ViraMiner.

## Supporting information

**S1 Table. Most common HHMER-identified viral proteins.** For the most commonly found viral proteins, the table shows how many 300 bp sequences were found in the entire dataset (train+val+test) and same count for only val+test sets. The measures in the last 3 columns are calculated using a combination of validation and test set, validation set was included to increase the number of samples (for more reliable measures). Average score corresponds to the model's mean output on the given (val+test) sequences. The quartiles show how these scores are distributed. The AUROC is calculated using all test-set non-viral sequences and only the val+test viral sequences corresponding to the given row. For comparison, the baseline values—using all test+val viral sequences, containing genes or not—are 0.390 for average score, [0.016; 0.127; 0.915] for quartiles and 0.923 for AUROC. It can be concluded that the model achieves improved performance on sequences containing these commonly found genes.
(PDF)

**S2 Table. Viral classes in the entire dataset.** The first column shows the viral classes (families) found in the dataset, the second column represents number of viral contigs identified by Blast.

The sequences are cut into 300 bp long sequences and the third column counts the numbers after the cut. Cutting longer contigs into smaller pieces means that the resulting 300 bp training sequences represent different parts of the same virus. "Others", at the last row of table, includes sequences that have by Blast been classified as definitely being viral, but have not been assigned a viral family yet.
(PDF)

## Acknowledgments

Supported by the Nordic Information for Action eScience Center awarded by the Nordic Academy for Advanced Sciences (NordForsk), the Swedish Research Council and the Swedish Foundation for Strategic Research.

## Author Contributions

**Conceptualization:** Ardi Tampuu, Zurab Bzhalava, Raul Vicente.

**Data curation:** Ardi Tampuu, Zurab Bzhalava.

**Formal analysis:** Ardi Tampuu, Zurab Bzhalava.

**Funding acquisition:** Joakim Dillner, Raul Vicente.

**Investigation:** Ardi Tampuu, Zurab Bzhalava.

**Methodology:** Ardi Tampuu, Zurab Bzhalava.

**Project administration:** Joakim Dillner, Raul Vicente.

**Resources:** Joakim Dillner, Raul Vicente.

**Software:** Ardi Tampuu, Zurab Bzhalava.

**Supervision:** Joakim Dillner, Raul Vicente.

**Validation:** Ardi Tampuu, Zurab Bzhalava, Joakim Dillner, Raul Vicente.

**Visualization:** Ardi Tampuu, Zurab Bzhalava.

**Writing – original draft:** Ardi Tampuu, Zurab Bzhalava, Raul Vicente.

**Writing – review & editing:** Ardi Tampuu, Zurab Bzhalava, Joakim Dillner.

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
