## [Decision Letter · Decision Letter 0]

4 Jul 2019

PONE-D-19-16058

ViraMiner: deep learning on raw DNA sequences for identifying viral genomes in human samples

PLOS ONE

Dear Mr Tampuu,

Thank you for submitting your manuscript to PLOS ONE. After careful consideration, we feel that it has merit but does not fully meet PLOS ONE’s publication criteria as it currently stands. Therefore, we invite you to submit a revised version of the manuscript that addresses the points raised during the review process.

**Both reviewers and myself are convinced that the investigators’ report a until now poorly examined avenue to exploration of metagenomic datasets for the presence of viral sequences. I anticipate that improvements, along suggested lines, can be carried out without further review.**

We would appreciate receiving your revised manuscript by Aug 18 2019 11:59PM. To enhance the reproducibility of your results, we recommend that if applicable you deposit your laboratory protocols in protocols.io, where a protocol can be assigned its own identifier (DOI) such that it can be cited independently in the future. For instructions see: http://journals.plos.org/plosone/s/submission-guidelines#loc-laboratory-protocols

We look forward to receiving your revised manuscript.

Kind regards,

Ulrich Melcher

Academic Editor

PLOS ONE

**Journal Requirements:**

**Additional Editor Comments (if provided):**

Language improvements needed:

, in occurrence order are listed here.

Line 2 “reside in” better than “reside on”? Or maybe both (in and on)

l. 10 delete “the”

l. 14 word order better: “obtain directly”

L.22 delete “percentage”?

l.. 30 vague antecedent: “it”

l. 57 “in” in place of “on” in description of the Figure Also line 80, line 290, 300, 305, 306

I would appreciate more explanation in a legend to Figure 1 how values for pattern branch and frequency branch are deduced.. Also: what is meant by “fully connected”?

l. 58 I prefer “puts out” and elsewhere

l. 80 number disagreement “these type”; also l. 309 “a…models’

l. 160 “We repeated out to…” ??? meaning not clear

l. 183-4 needs articles; ‘the former, ‘the latter’.

L. 210-211 is not clear. Rewrite

l. 262-3 needs work

l.318-321 is not clear to me

**Comments to the Author**

1. Is the manuscript technically sound, and do the data support the conclusions?

Reviewer #1: Yes

Reviewer #2: Yes

2. Has the statistical analysis been performed appropriately and rigorously? 

Reviewer #1: Yes

Reviewer #2: Yes

3. Have the authors made all data underlying the findings in their manuscript fully available?

Reviewer #1: Yes

Reviewer #2: Yes

4. Is the manuscript presented in an intelligible fashion and written in standard English?

Reviewer #1: Yes

Reviewer #2: Yes

5. Review Comments to the Author

Reviewer #1: This is a very well written manuscript with promising results of 19 human metagenomes, however, there are some items that I considered they need to be addressed before publication:

Although the machine learning model shows promising results, it would be interesting to the reader to know which genome regions of the viruses were used by the trained model to identify the potential new viruses. Is there any bias towards coat protein, RNA dependent RNA polymerase or any other genes to determine the presence of the virus in the dataset?

Can you include in the results what are the hallmark genes that were mostly found in the identified sequences? This would possibly require HMMER to be run using the unknown sequences of your 19 metagenomes after blastn.

Why didn’t you try contig lengths larger than 300, for instance 1000, 5000, 10000? Would your model perform better with those contig lengths?

The tool requires further validation with more data. I understand that you are using 19 metagenomes and using partitions/baselines to train, and the AUROC can be considered a good parameter to evaluate your model. However, it is imperative to certainly know what is in your metagenome to be able to validate the current technology. I’d suggest to generate simulated human metagenomes using taxon profiles similar to the ones that you used in the training model. I would suggest using NeSSM, ART, MetaSim for the simulations to determine how your trained model performs in completely new datasets.

Per line comments:

Line 40: Builds on top of the CNN architecture of Ren [25].

Line 101: Spell out AUROC as Area Under the Receiver Operating Characteristics the first time in the text.

Line 185: You can say something like: “trained to identify viruses infecting prokaryotic organisms”

Line 251: replace producing by produced.

Reviewer #2: This article proposed a new machine learning approach for characterizing unknown metagenomics contigs. The approach using ANN with raw DNA sequences as inputs is unique and novel. The authors demonstrated that the proposed approach “viralMiner” performs better than random forests and kmer as baseline. The writing is excellent as well. Because of the novelty of the approach, I recommend the paper accepted after minor revision.

Several minor areas can be imporved:

1. The AUC is 0.92, however, the real performance 0.9 accuracy and 0.32 recall is not as impressive. I believe these numbers are much worse than blast, so I recommend emphasize this in the abstract

2. The real strength of this approach is to detect “unknown” contigs which cannot be detected by blast. However the training test validation experiments did not evaluate anything that is “unknown”. Maybe the authors can hold out some viral classes in the training and test if the machine learning approach can detect “unknown” contigs?

3. A table could be added to show all training viral classes.

6. PLOS authors have the option to publish the peer review history of their article (what does this mean?). If published, this will include your full peer review and any attached files.

Reviewer #1: No

Reviewer #2: No

---

## [Author Response · Author response to Decision Letter 0]

7 Aug 2019

Responses to additional Editor Comments

Language improvements needed:

, in occurrence order are listed here.

Line 2 “reside in” better than “reside on”? Or maybe both (in and on) 

Reply: Changed as requested to “in and on”

l. 10 delete “the” 

Reply: Changed as requested

l. 14 word order better: “obtain directly” 

Reply: Changed as requested

L.22 delete “percentage”? 

Reply: Changed as requested

l.. 30 vague antecedent: “it” 

Reply: Added “the algorithm” in the text instead of “it”

l. 57 “in” in place of “on” in description of the Figure Also line 80, line 290, 300, 305, 306 

Reply: Changed as requested

I would appreciate more explanation in a legend to Figure 1 how values for pattern branch and frequency branch are deduced.

Also: what is meant by “fully connected”? 

Reply: We have expanded the caption of Figure 1 and explicitly clarified the meaning of “fully-connected” both in figure caption and the text. 

l. 58 I prefer “puts out” and elsewhere

Reply: In the context of computer science, “to output” is more commonly used than “to put out” in the machine learning community. In fact, Google Scholar searches for “the model outputs a” and “model puts out a” show that the first phrasing (using “to output”) is much more common. The exact sentence “the model outputs a” was used to i) to avoid responses where “outputs” is a noun and ii) to minimize confounding results due to Google not paying attention to punctuation and capitalization -- results such as “the model outputs. The”. Thus, using “to output” seems grammatically valid, widely-used, and we would prefer to use it in the context of our manuscript. 

l. 80 number disagreement “these type”; also l. 309 “a…models’ 

Reply: Changed as requested

l. 160 “We repeated out to…” ??? meaning not clear 

Reply: Changed, the original sentence was indeed nonsensical with some excessive or missing part. 

l. 183-4 needs articles; ‘the former, ‘the latter’. 

Reply: Changed as requested

L. 210-211 is not clear. Rewrite

Reply: The sentence was rewritten as “However, it is important to note that ViraMiner does not have a database to rely on and must thus capture a different kind of information of genome composition than alignment based methods.”

l. 262-3 needs work 

Reply: For clarity, we have now added a few sentences that put convolutional networks in context by mentioning their particularities. Also, we expand on the learning procedure. Notice that the nature of convolutions in case of 1D input is explained later in the subsection. 

l.318-321 is not clear to me

Reply: This has been expanded in Methods and Materials

Response to Reviewers

Reviewer #1: 

This is a very well written manuscript with promising results of 19 human metagenomes, however, there are some items that I considered they need to be addressed before publication:

1) Although the machine learning model shows promising results, it would be interesting to the reader to know which genome regions of the viruses were used by the trained model to identify the potential new viruses. Is there any bias towards coat protein, RNA dependent RNA polymerase or any other genes to determine the presence of the virus in the dataset?

Can you include in the results what are the hallmark genes that were mostly found in the identified sequences? This would possibly require HMMER to be run using the unknown sequences of your 19 metagenomes after blastn.

Reply: We thank the reviewer for these comments. We ran HMMER with pfam database on the viral 300bp sequences and the table below represents the found proteins. The table only contains the most frequent proteins, because we want to have at least 10 data points when calculating average metrics. The second column counts the occurrences of the protein in the entire dataset (train+val+test). The third column counts the occurrences of the protein among val+test set samples. The fourth column shows the average probability of being viral according to the model for val+test set samples of the given protein (i.e. the average output value). The fifth column gives the quartiles of these score values. The sixth column gives the AUROC (using test+val sequences containing given protein and all test set non-viruses). 

There are clear differences among the values in the fourth, fifth and sixth column. More importantly, all these values are an improvement over the average performance over all viral sequences (given in table caption). Hence, indeed, containing parts of certain proteins helps the model to identify the sequence as viral. 

This table is now provided as supplementary table for the paper. 

S1 Table. Most common HHMER-identified viral proteins. For the most commonly found viral proteins, the table shows how many 300bp sequences were found in the entire dataset (train+val+test) and same count for only val+test sets. The measures in the last 3 columns are calculated using a combination of validation and test set, validation set was included to increase the number of samples (for more reliable measures). Average score corresponds to the model’s mean output on the given (val+test) sequences. The quartiles show how these scores are distributed. The AUROC is calculated using all test-set non-viral sequences and only the val+test viral sequences corresponding to the given row. For comparison, the baseline values - using all test+val viral sequences, containing genes or not - is 0.390 for average score, [0.016; 0.127; 0.915] for quartiles and 0.923 for AUROC. It can be concluded that the model achieves improved performance on sequences containing these commonly found genes.

----------------

2) Why didn’t you try contig lengths larger than 300, for instance 1000, 5000, 10000? Would your model perform better with those contig lengths?

Reply: As mentioned in “Data processing and labeling” subsection, we also tried sequence length 500, but it performed clearly worse than 300. In initial experiments we also briefly tested sequence length 1000, but the results were even weaker. We hypothesize that this is due to having less data points with longer (more restrictive) sequence lengths. 

With sequence length 500 we have 3 times less data than with sequence length 300. With sequence length 1000 we have 20 times less data than with length 300. Empirical results show that in the current case having more data is the most important.

-----------------

3) The tool requires further validation with more data. I understand that you are using 19 metagenomes and using partitions/baselines to train, and the AUROC can be considered a good parameter to evaluate your model. However, it is imperative to certainly know what is in your metagenome to be able to validate the current technology. I’d suggest to generate simulated human metagenomes using taxon profiles similar to the ones that you used in the training model. I would suggest using NeSSM, ART, MetaSim for the simulations to determine how your trained model performs in completely new datasets.

Reply: We believe that the true effectiveness of our models can only be measured by their performance on real data, preferably originating from a sequencing experiment that has not been used for training the model. Leaving an entire dataset out from the training procedure (using training set) and model picking (using validation data) procedure achieves just that.

That said, we agree that working with real data, there is always some risk of unknown biases making results nicer than they should be. We have thus repeated our experiments on simulated data, as requested. 

We considered all three simulation tools mentioned by the reviewer and decided to use ART, as it is most understandable and easy to use. The article now contains a new methods section describing the simulation procedure and parameters, and a results subsection describing the results obtained.

The results showed that the model that was trained our 19 metagenomic experiments produced AUROC 0.751 on the simulated data. Even though the model performs clearly above the random level, it is indeed a more moderate performance compared to the model performance on the main test set. However, consider that the simulated dataset was generated based on randomly picked viral reference genomes from GenBank without any prior selection.

Using a ViraMiner model both trained and tested on simulated data, the test AUROC increases to 0.921. We believe that this further proves that the architecture is useful and able to generalize to different datasets.

---------------

Per line comments:

Line 40: Builds on top of the CNN architecture of Ren [25]. 

Reply: Changed as requested 

Line 101: Spell out AUROC as Area Under the Receiver Operating Characteristics the first time in the text. 

Reply: Changed as requested 

Line 185: You can say something like: “trained to identify viruses infecting prokaryotic 

organisms” 

Reply: We thank the reviewer for pointing this out. Changed as requested 

Line 251: replace producing by produced. 

Reply: Changed as requested 

Reviewer #2: 

This article proposed a new machine learning approach for characterizing unknown metagenomics contigs. The approach using ANN with raw DNA sequences as inputs is unique and novel. The authors demonstrated that the proposed approach “viralMiner” performs better than random forests and kmer as baseline. The writing is excellent as well. Because of the novelty of the approach, I recommend the paper accepted after minor revision.

Several minor areas can be improved:

1) The AUC is 0.92, however, the real performance 0.9 accuracy and 0.32 recall is not as impressive. I believe these numbers are much worse than blast, so I recommend emphasize this in the abstract.

Reply: 

Blast was used as reference method (i.e. always correct by definition) and a comparison of performance with the reference method is not useful. In the abstract, we suggest to use our method only after having applied conventional alignment methods (BLAST) in order to further investigate the sequences that are left unlabeled. 

Also, as seen from precision-recall curve, instead of precision-recall pair 0.9&0.32, any other pair (e.g. 0.95&0.24 or 1.0&0.151) could have been chosen. It is not possible to foresee which particular precision-recall ratio is the most important to the reader. Hence, we are reluctant to stress one particular pair of values by mentioning them in the abstract.

In this manuscript we have decided to use AUROC as the main metric, because it does not depend neither on class distribution nor on a particular classification threshold. 

 ---------

2) The real strength of this approach is to detect “unknown” contigs which cannot be detected by blast. However the training test validation experiments did not evaluate anything that is “unknown”. Maybe the authors can hold out some viral classes in the training and test if the machine learning approach can detect “unknown” contigs?

Reply: We thank the reviewer for this comment. The primary purpose use case of ViraMiner is to identify distant homologs that alignment-based methods cannot detect reliably. To maximally detect those highly divergent viruses, we think that all viral classes should be included in the training dataset. In comparison, detecting new yet unknown viral classes is a much more complicated task since one viral family can be very different from the others. Note that even though the CNN-based method does not use the same type of similarity measures as BLAST or HMMER3, it still must rely on some kind of common features. 

Nevertheless, we have now investigated if ViraMiner is able to identify viruses from a viral class that it had not been trained on. For this purpose, we trained and validated the ViraMiner model on a dataset where all anelloviruses were removed. We used the same hyperparameters (layer sizes) as above, with the same stepwise training strategy - i.e. no new hyperparameter search was performed. The test set contained non-viral samples (10% of all non-viruses) and the left-out anelloviruses. 

In this setting, from the model's point of view anelloviruses are a completely unknown and unseen type of viruses. On this test set, frequency branch alone achieves the best result, with 0.755 AUROC. This is clearly above random performance and translates into getting 11 of the top 20 predictions correct on a test set with 5% prevalence. We conclude that even when dealing with viral sequences that are distant from our training (and validation) samples, using ViraMiner as recommendation system increases the chances of identifying viruses.

-----------

3) A table could be added to show all training viral classes.

Reply: Such table is now provided in the manuscript.

S2 Table. Viral classes in the entire dataset. The first column shows the viral classes (families) found in the dataset, the second column represents number of viruses found by Blast. The sequences are cut into 300 bp long sequences and the third column counts the numbers after the cut. Cutting longer contigs into smaller pieces means that the resulting 300bp training sequences represent different parts of the same virus. “Others”, at the last row of table, includes sequences that have by Blast been classified as definitely being viral, but have not been assigned a viral family yet.

---

## [Editor Report · Decision Letter 1]

27 Aug 2019

ViraMiner: deep learning on raw DNA sequences for identifying viral genomes in human samples

PONE-D-19-16058R1

Dear Dr. Tampuu,

We are pleased to inform you that your manuscript has been judged scientifically suitable for publication and will be formally accepted for publication once it complies with all outstanding technical requirements.

With kind regards,

Ulrich Melcher

Academic Editor

PLOS ONE

Additional Editor Comments (optional):

I apologize for the various delays and misunderstandings.  As recognized during the first reading, the submission showed great promise, but also some language work.  The current revision is much more comprehensible.  I will leave it up to the authors to act on my suggestions for this version.

Line 130

In here, we exemplify the use of ViraMiner a recommendation system.

Better?:

Herein, we exemplify the use of ViraMiner as a recommendation system.

152

because among our datasets largest number

better?:

because, among our datasets, the largest number

l.174

In addition, we investigated if ViraMiner

In addition, we investigated whether ViraMiner

l. 236  number disagreement

"The most important criteria that ViraMiner had to satisfy, however, were"

OR use plural "Criterion"

L. 416 add "the"

In the  results section

---

## [Editor Report · Acceptance letter]

3 Sep 2019

PONE-D-19-16058R1 

ViraMiner: deep learning on raw DNA sequences for identifying viral genomes in human samples 

Dear Dr. Tampuu:

I am pleased to inform you that your manuscript has been deemed suitable for publication in PLOS ONE. Congratulations! Your manuscript is now with our production department. 

With kind regards,

on behalf of

Dr. Ulrich Melcher 

Academic Editor

PLOS ONE